# Medical care needs for patients receiving home healthcare in Taiwan: Do gender and income matter?

**Fang-Yi Huang[1], Chung-Han Ho[2,3,4], Jung-Yu Liao[5], Chao A. Hsiung[6], Sang-Ju Yu[7], Kai-Ping Zhang[8], Ping-Jen Chen[9,10,11]** *

1 Department of Social and Policy Sciences, Yuan Ze University, Taoyuan, Taiwan, 2 Department of Medical Research, Chi-Mei Medical Center, Tainan, Taiwan, 3 Department of Hospital and Health Care Administration, Chia Nan University of Pharmacy and Science, Tainan, Taiwan, 4 Cancer Center, Wan Fang Hospital, Taipei Medical University, Taipei, Taiwan, 5 Department of Public Health, Kaohsiung Medical University, Kaohsiung, Taiwan, 6 Institute of Population Health Sciences, National Health Research Institutes, Miaoli, Taiwan, 7 Home Clinic Dulan, Taitung, Taiwan, 8 Home Clinic Dulan, Taipei, Taiwan, 9 Marie Curie Palliative Care Research Department, Division of Psychiatry, University College London, London, United Kingdom, 10 Department of Family Medicine and Division of Geriatrics and Gerontology, Kaohsiung Medical University Hospital, Kaohsiung Medical University, Kaohsiung, Taiwan, 11 School of Medicine, Kaohsiung Medical University, Kaohsiung, Taiwan

☯ These authors contributed equally to this work.
* pingjen.chen@gmail.com

**Data Availability Statement:** The data for this study was obtained from the National Health Insurance Research Database (NHIRD, http://nhird.nhri.org.tw/). The application for permission to

## Abstract

Studies about medical care needs for home healthcare (HHC) previously focused on disease patterns but not gender and income differences. We used the Taiwan National Health Research Insurance Database from 1997 to 2013 to examine trends in medical care needs for patients who received HHC, and the gender and income gaps in medical care needs, which were represented by resource utilization groups (RUG). We aimed to clarify three questions: 1. Are women at a higher level of medical care needs for HHC than men, 2. Does income relate to medical care needs? 3. Is the interaction term (gender and income) related to the likelihood of medical care needs? Results showed that the highest level of medical care need in HHC was reducing whereas the basic levels of medical care need for HHC are climbing over time in Taiwan during 1998 and 2013. The percentages of women with income-dependent status in RUG1 to RUG4 are 26.43%, 26.24%, 30.68%, and 32.07%, respectively. Women were more likely to have higher medical care needs than men (RUG 3: odds ratio, OR = 1.17, 95% confidence interval, CI = 1.10–1.25; RUG4: OR = 1.13, 95% CI = 1.06–1.22) in multivariates regression test. Compared to the patients with the high-income status, patients with the income-dependent status were more likely to receive RUG3 (OR = 2.34, 95% CI = 1.77–3.09) and RUG4 (OR = 1.98, 95% CI = 1.44–2.71). The results are consistent with the perspectives of fundamental causes of disease and feminization of poverty theory, implying gender and income inequalities in medical care needs. Policymakers should increase public spending for delivering home-based integrated care resources, especially for women with lower income, to reduce the double burden of female poverty at the higher levels of medical care needs for HHC.

access the data was sent from Chi-Mei Medical Center to National Health Research Institutes (NHRI), Taiwan and approved. But restrictions apply to the availability of these data, which were used under license for the Chi-Mei Medical Center and current study only, and so are not publicly available. Data are however can be checked for any researcher who may concern about its reliability upon reasonable request to the Department of Research or Institutional Review Board in Chi-Mei Medical Center, Taiwan (https://www.chimei.org. tw/main/cmh_department/top/54000_index.html). For further analyses of these data, researchers should apply for permission independently from the NHIRD, which has been transferred to the Health and Welfare Data Science Center (HWDC), Department of Statistics, Ministry of Health and Welfare, Taiwan (http://dep.mohw.gov.tw/DOS/np-2497-113.html).

**Funding:** The project was funded by the Chi-Mei Medical Center (grant reference: CMHCR11009), and Ping-Jen Chen was supported by a Government Scholarship for Overseas PhD Study, Ministry of Education, Taiwan (grant reference: 1051165-1-UK-002). The funders had no role in study design, data collection and analysis, decision to publish, or preparation of the manuscript.

**Competing interests:** The authors have declared that no competing interests exist.

# Introduction

## Home healthcare and medical care needs

The needs for home healthcare (HHC) is increasing due to the aim of aging in place in the era of global population aging. Compared to a larger literature on the effects of social determinants on medical care needs [1–5], however, fewer have concentrated on HHC patients who live at home. Some studies investigated the association between the use of home health services, the access of device use [5], the number of readmissions [6], and the duration of rehabilitation care [7]. Little was known about specialized medical care needs such as tracheostomy and urinal catheterization care, intravenous injection, or colostomy irrigation in the HHC recipients.

## Gender and medical care needs for HHC

Admittedly, HHC has undergone enormous growth over the last 30 years but studies about the relationships between gender and medical care needs for HHC service are inconclusive. Some studies found that women are more likely to receive HHC and report greater unmet home care needs than men [8, 9], and this gender gap increases by age [10]. Yet, other research reported that men presented with higher levels of need for HHC since men had higher rates of most chronic conditions, limitations in activities of daily living, and instrumental activities of daily living than women [11]. Moreover, in the patients who received HHC services, men have higher frequencies of medical care utilization than women among the disabled group [12] and also were more likely to have multiple hospitalizations in the last 3 months of life than women in people with dementia [13].

## Socioeconomic status and medical care needs

The literature confirms that poverty and the level of urbanization are linked with medical care needs [14]. In general, patients with lower socioeconomic status have a lower level of medical care utilization but higher unmet needs in HHC than those with higher socioeconomic status. For instance, Saeed et al. [1] found that patients with lower incomes and living in rural area have less medical care use due to the difficulty of financial accessibility. Forbes et al. [9] also found that patients with lower socioeconomic status had more unmet home care needs than those with higher socioeconomic status since support services of HHC often are expensive, especially for people with dementia.

However, others found that socioeconomic status is not related to medical care needs. For example, Freedman et al. [5] argued that Medicare managed care enrollees with low socioeconomic status have higher access to HHC visits. Still, the minority with low income like African Americans have a greater resource utilization and greater reimbursement of home health services with increased risk for morbidity than the richer white [15]. Thus, whether the lower socioeconomic status is associated with medical care needs remaining uncertain.

## Theory and arguments

Guided by the perspective of "fundamental causes of health and disease", poorer socioeconomic status is the fundamental cause of diseases [16]. Since 1995, Bruce Link and Jo Phelan have confirmed that social conditions cause illness through two pathways. One is those with low socioeconomic status are less likely to have access to health care [17]. The other is that the mechanisms of risky health behaviors such as poor diet, stressful life, heavy smoking, less social support, and higher job hazards could lead to more anxiety, depression, higher cholesterol, blood pressure, and heart disease [18, 19]. According to the perspective of the feminization of poverty, the inequality in living standards between men and women is increasing at old age,

and the majority of old women face the highest risk of poverty due to lack of income and resources [20]. Based on the theories of fundamental causes of disease and feminization of poverty [16, 20], we argued that women, patients with lower income, and those who live in more rural areas may be more likely to have a higher level of medical care needs. Hypotheses are as follows:

1. Women are more likely to have higher levels of medical care needs in HHC than men.

2. Lower income is more likely to be associated with higher levels of medical care needs in HHC compared to the higher income group.

3. Women who are in income-dependent or low-income groups are more likely to be associated with higher levels of medical care needs in HHC than those with moderate income and high income.

## Material and methods

### Data sources and samples

The data for this cross-sectional population-based study was obtained from the National Health Insurance Research Database (NHIRD) from 1997 to 2013 in Taiwan. NHIRD is a nationwide database which contains the data of more than 23 million individuals and is managed by the National Health Research Institute (NHRI) before 2015. National Health Insurance (NHI) system, which is a single-payer universal healthcare scheme and covers 99% of the population in Taiwan [21], has been launched in 1995 [22]. We conducted a population-based study to examine the relationship between social determinants, diseases, and medical care needs levels in HHC patients. We included all patients who received HHC from 60 to 105 years old. The study was approved by the Institutional Review Board of Chi Mei Medical Center (IRB No. 10410-E01), and the requirement for informed consent was waived.

### Outcome variables and covariates

The medical care needs of patients who received HHC in Taiwan are classified by resource utilization groups (RUGs) from level 1 to level 4, and HHC services at different levels of RUGs are all reimbursed with correlated payments by NHI. The first level RUG is intended for those who only require ordinary healthcare at home and level 2 to 4 of RUG are aimed at people who require one, two, three, or more specialized care services respectively. specialized care services include "change tracheostomy set", "urinal indwelling catheterization", "insertion of nasogastric tube", "bladder irrigation", "wound treatment", "intravenous drip", and "colostomy irrigation". Medical care need is a nominal variable.

According to a behavioral model for explaining health care utilization developed by Andersen and Newman [23], we categorized covariates into three types, including predisposing characteristics (age and gender), enabling characteristics (income and urbanization), and need characteristics (major diseases and comorbidity). The enabling characteristics reflect the location (status) of the individuals in their society as measured by income, education, and urbanization levels of the residence. In our study, income was divided into four levels: one is the dependent group, who are usually wives, children, parents and their health insurance covered by their spouses or relatives. The other three levels of income were defined by salary-based health insurance monthly premiums, (1) low: lower than US$571 per month (New Taiwan Dollar (NT$) 20000); (2) moderate: between US$571–1141 per month (NT$20000–40000); and (3) high: US$1142 per month (NT$40001) or more. This study regarded income as one proxy of individual socioeconomic status.

Another enabling characteristic was the urbanization level of residence. The urbanization level of residence was classified into four levels of residence from urban to rural areas based on 5 indices in Taiwan: population density, percentage of residents with college-level or higher education, percentage of residents > 65 years old, percentage of residents who are agriculture workers, and the number of physicians per 100000 patients [24].

The need characteristics include comorbidities defined by the International Classification of Diseases, Ninth Revision, Clinical Modification (ICD-9-CM), such as cancer, neurodegenerative diseases (e.g. dementia or parkinsonism), stroke, heart failure, chronic obstructive pulmonary disease, chronic liver disease and cirrhosis, chronic kidney disease, hypertension, diabetes, coronary artery disease, hyperlipidemia, atrial fibrillation, tuberculosis, and epilepsy.

## Statistical analysis

Descriptive statistics were conducted to display the distribution of the studying variables in the population sample in Table 1. Fig 1 was used to present how medical care needs changing from 1997 to 2013. In addition, $\chi^2$ analysis for categorical variables and analysis of variance (ANOVA) test for the continuous variable (age) were used to examine the differences between each independent variable and the level of RUGs. Because the RUGs were grouped into more than two categories, we adopted multinomial logistic regression to estimate the associations between variables, eight gender-income groups (an interaction term of gender multiplying income), and medical care needs as in Table 4 and Table 5. The analysis was also done for patients with cancer and stroke specifically due to their lowest and highest prevalence in major diseases served by HHC. All the analysis was conducted by STATA 13 and incorporated the weighted procedure used in the NHIRD sampling design.

## Results

### Change of medical care needs in HHC patients over time

As shown in Fig 1, there are more and more HHC patients who have been in the second level of RUG from 1997 to 2013. The percentage of patients in RUG4 kept decreasing from 1999, while the proportion of patients with higher medical care needs (RUG 3 and RUG4) got steady during 1998 and 2002 and started to decline since 2003.

### Characteristics of HHC patients

Table 1 showed patient characteristics. The percentages of the four levels of medical care need among HHC patients are 1.89%, 49.02%, 41.02%, and 8.07% respectively. The majority of patients are with the first and second levels of income: "the dependent group" (47.62%) and "low" (33.70%). Most respondents resided in very urban or urban areas. Among all HHC patients, 16.91% had cancer, 31.29% got neurodegenerative diseases, 70.15% had a stroke, 25.58% had heart failure, 42.76% got chronic obstructive pulmonary disease, 19.27% had chronic liver disease and cirrhosis, and 25.58% of the population had chronic kidney disease. These HHC patients have the highest prevalence of hypertension (81.03%) and the lowest prevalence of epilepsy (6.36%).

### Variables related to medical care needs in HHC

Distributions in various variables and comorbidities for patients receiving home healthcare at different medical care need levels are shown in Table 2. The proportion of women are higher than men in groups with higher medical care needs (54.1% versus 45.9% in RUG3, 50.2%

**Table 1. Characteristics for home healthcare patients in Taiwan from 1997 to 2013 (N = 238,176).**

| Variables | Category | Frequency | Distribution (%) |
|---|---|---|---|
| **Medical Care Needs** | RUG 1 | 4,491 | 1.89 |
| | RUG 2 | 116,753 | 49.02 |
| | RUG 3 | 97,710 | 41.02 |
| | RUG 4 | 19,222 | 8.07 |
| **Age** [a] | | 79.47 [b] | 8.37 [b] |
| **Gender** | Male | 116,547 | 48.93 |
| | Female | 121,629 | 51.07 |
| **Income** [c] | Dependent | 113,420 | 47.62 |
| | Low | 80,276 | 33.70 |
| | Moderate | 43,172 | 18.13 |
| | High | 1,308 | 0.55 |
| **Urbanization** | Urban | 122,416 | 51.40 |
| | Sub-urban | 91,055 | 38.23 |
| | Sub-rural | 21,018 | 8.82 |
| | Rural | 3,687 | 1.55 |
| **Comorbidities** | | | |
| **Cancer** | | 40,266 | 16.91 |
| **Neurodegenerative diseases** | | 74,520 | 31.29 |
| **Stroke** | | 167,070 | 70.15 |
| **Heart failure** | | 60,935 | 25.58 |
| **COPD** | | 101,848 | 42.76 |
| **Chronic liver disease** | | 45,896 | 19.27 |
| **Chronic kidney disease** | | 43,613 | 18.31 |
| **Hypertension** | | 192,983 | 81.03 |
| **Diabetes** | | 27,443 | 11.52 |
| **Coronary artery disease** | | 108,066 | 45.37 |
| **Hyperlipidemia** | | 72,766 | 30.55 |
| **Atrial fibrillation** | | 32,883 | 13.81 |
| **Tuberculosis** | | 16,400 | 6.89 |
| **Epilepsy** | | 15,157 | 6.36 |

COPD = chronic obstructive pulmonary disease, RUG = resource utilization groups.

[a] Age range is from age 60 to age 105.

[b] The mean (standard deviation) are shown for the variables: age.

[c] Income dependent group refers to people whose health insurance of premiums were covered by their family members who have income. And the other three income levels were defined by salary-based health insurance premiums.

versus 49.8% in RUG4), whereas the trend of gender ratio reversed in groups with lower medical care needs. In the group with the highest level of medical care needs, the percentage of people with income-dependent status (54%) is much higher than their counterparts in other medical care needs groups.

Trends of the proportion of each medical care need group from RUG1 to RUG4 are dissimilar by the types of diseases. For instance, cancer patients and stroke patients had the lowest and highest percentages in RUG3 (14.93% versus 72.65%) and RUG4 (12.26% versus 75.41%) respectively. In addition, the percentages of cancer patients are decreasing from RUG1 to RUG4 groups (23.20%, 19.08%, 14.93%, and 12.26%) whereas the proportions of stroke patients are increasing (50.86%, 67.93%, 72.65%, and 75.41%) according to Table 2.

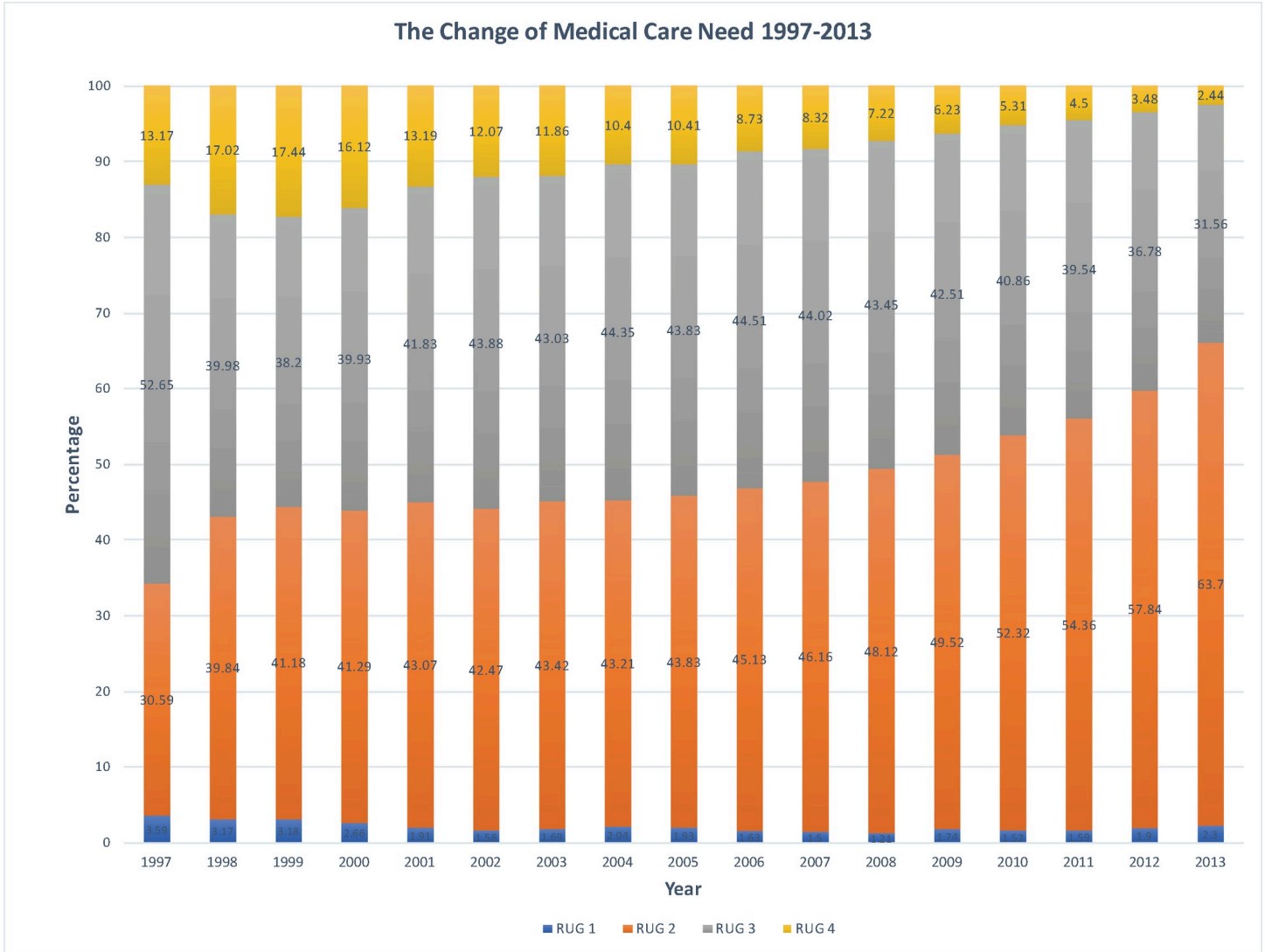

**Fig 1. The change in the proportion of medical care need levels during 1997–2013.** Note: 1. The total number of cases is 238,176. 2. RUG = resource utilization group.

### The relationships between gender, income, and medical care needs

Table 3 displays the association between gender-income groups and medical care need levels among all study subjects, cancer patients, and stroke patients. We found out there is a significant difference in proportions of gender-income groups among four medical care need levels of HHC. Overall, women with income-dependent status are more likely to have higher medical care needs compared to other gender-income groups. For example, the percentages of women with income-dependent status in RUG1 to RUG4 groups are 26.43%, 26.24%, 30.68%, and 32.07% respectively. At the same time, the percentages of women with high income in RUG1 to RUG4 groups are 0.38%, 0.14%, 0.10%, and 0.15%.

### Multinominal regression analysis for medical care needs

In Table 4, we found that women were more likely to receive RUG 3 (odds ratio, OR = 1.17, 95% confidence interval, CI = 1.10–1.25) and RUG4 (OR = 1.13, 95% CI = 1.06–1.22) than

**Table 2. Distribution in age, gender, socioeconomic variables, and comorbidities for patients receiving home healthcare at different medical care need levels.**

| | RUG 1 (n = 4,491) | RUG 2 (n = 116,753) | RUG 3 (n = 97,710) | RUG 4 (n = 19,222) |
|---|---|---|---|---|
| **Age** (year, mean ± SD) | 78.08 ±0.13 | 79.96±0.02 | 79.48 ±0.03 | 76.75± 0.06 |
| **Gender** | | | | |
| Male (%) | 51.48 | 51.23 | 45.91 | 49.75 |
| Female (%) | 48.52 | 48.77 | 54.09 | 50.25 |
| **Income**[a] | | | | |
| Dependent (%) | 45.91 | 45.48 | 48.99 | 54.03 |
| Low (%) | 35.94 | 33.83 | 33.16 | 35.22 |
| Moderate (%) | 16.81 | 20.10 | 17.42 | 10.07 |
| High (%) | 1.34 | 0.60 | 0.43 | 0.68 |
| **Urbanization** | | | | |
| Urban (%) | 52.15 | 51.66 | 50.99 | 51.69 |
| Sub-urban (%) | 38.88 | 38.10 | 38.32 | 38.42 |
| Sub-rura (%) | 7.59 | 8.70 | 9.09 | 8.50 |
| Rural (%) | 1.38 | 1.54 | 1.60 | 1.39 |
| **Comorbidities** | | | | |
| Cancer (%) | 23.20 | 19.08 | 14.93 | 12.26 |
| Neurodegenerative diseases (%) | 18.62 | 32.27 | 31.82 | 25.60 |
| Stroke (%) | 50.86 | 67.93 | 72.65 | 75.41 |
| Heart failure (%) | 33.80 | 26.79 | 24.63 | 21.19 |
| COPD (%) | 49.77 | 43.57 | 41.78 | 41.18 |
| Chronic liver disease (%) | 20.82 | 20.34 | 18.58 | 15.89 |
| Chronic kidney disease (%) | 18.39 | 19.85 | 17.32 | 13.97 |
| Hypertension (%) | 75.40 | 81.84 | 81.04 | 77.32 |
| Diabetes (%) | 10.75 | 11.26 | 11.93 | 11.21 |
| Coronary artery disease (%) | 45.94 | 47.03 | 44.64 | 38.91 |
| Hyperlipidemia (%) | 31.08 | 31.98 | 30.14 | 23.86 |
| Atrial fibrillation (%) | 15.34 | 14.20 | 13.51 | 12.58 |
| Tuberculosis (%) | 10.47 | 7.33 | 6.29 | 6.41 |
| Epilepsy (%) | 4.05 | 5.98 | 6.61 | 8.02 |

COPD = chronic obstructive pulmonary disease, RUG = resource utilization groups, SD = standard deviation. All indicate the significance p< 0.001 by χ2 test. F value is 858.80 for ANOVA test between age and RUGs.

[a] Income dependent group refers to people whose health insurance of premiums were covered by their family members who have income. And the other three income levels were defined by salary-based health insurance premiums.

men. Compared to the patients with the high-income status, patients with the income-dependent status were more likely to receive RUG3 (OR = 2.34, 95% CI = 1.77–3.09) and RUG4 (OR = 1.98, 95% CI = 1.44–2.71). Patients with the low-income status (RUG3: OR = 2.18, 95% CI = 1.65–2.88; RUG4: OR = 1.81, 95% CI = 1.32–2.48) had similar patterns in medical care needs with the patients with income-dependent status (RUG3: OR = 2.34, 95% CI = 1.77–3.09; RUG4: OR = 1.98, 95% CI = 1.44–2.71) compared to high-income patients respectively. Moreover, the patterns of medical care needs in HHC differ by types of disease. Particularly speaking, cancer patients were less likely to have higher medical care needs, especially in RUG4 (OR = 0.58, 95% CI = 0.53–0.63). However, stroke patients were most likely to receive HHC in higher medical needs (RUG3: OR = 2.43, 95% CI = 2.27–2.59; RUG4: OR = 2.86, 95% CI = 2.66–3.08).

We further examined the associations between interaction terms of gender and income, and RUGs among all study subjects, cancer patients, and stroke patients. As shown in Table 5,

**Table 3. The relationships between gender, income, and medical care need levels in home healthcare among all study subjects, cancer patients, and stroke patients.**

| | All study subjects (N = 238,176) | | | | Cancer patients (N = 40,266) | | | | Stroke patients (N = 167,070) | | | |
|---|---|---|---|---|---|---|---|---|---|---|---|---|
| | RUG1 | RUG2 | RUG3 | RUG4 | RUG1 | RUG2 | RUG3 | RUG4 | RUG1 | RUG2 | RUG3 | RUG4 |
| **Male & Income dependent** (%) | 19.46 | 19.23 | 18.30 | 21.94 | 21.21 | 20.13 | 19.56 | 23.50 | 20.27 | 20.91 | 19.38 | 22.79 |
| **Female & Income dependent** (%) | 26.45 | 26.26 | 30.69 | 32.09 | 22.46 | 22.47 | 25.14 | 25.67 | 26.88 | 25.66 | 30.74 | 32.07 |
| **Male & Low income** (%) | 22.60 | 22.19 | 20.06 | 22.16 | 25.62 | 25.17 | 23.49 | 25.63 | 24.34 | 22.55 | 20.04 | 22.19 |
| **Female & Low income** (%) | 13.34 | 11.64 | 13.10 | 13.06 | 12.09 | 10.04 | 11.28 | 11.50 | 11.56 | 10.46 | 12.26 | 12.36 |
| **Male & Moderate income** (%) | 8.46 | 9.36 | 7.22 | 5.11 | 10.27 | 11.29 | 9.60 | 7.30 | 8.10 | 9.66 | 7.24 | 5.17 |
| **Female & Moderate income** (%) | 8.35 | 10.73 | 10.20 | 4.96 | 6.24 | 9.96 | 10.23 | 5.43 | 8.01 | 10.20 | 9.93 | 4.76 |
| **Male & High income** (%) | 0.96 | 0.45 | 0.33 | 0.54 | 1.44 | 0.78 | 0.57 | 0.85 | 0.57 | 0.44 | 0.33 | 0.51 |
| **Female & High income** (%) | 0.38 | 0.14 | 0.10 | 0.15 | 0.67 | 0.17 | 0.14 | 0.13 | 0.26 | 0.12 | 0.09 | 0.14 |

RUG = resource utilization groups, All results for χ2 tests between RUGs and eight gender-income groups are all with p values below 0.001. Income dependent group refers to people whose health insurance of premiums were covered by their family members who have income. And the other three income levels were defined by salary-based health insurance premiums.

among all study subjects, women with income-dependent status and low-income were more likely to receive RUG3 (OR = 2.45, 95% CI = 1.77–3.41; OR = 2.19, 95% CI = 1.57–3.05; respectively) and RUG4 (OR = 2.02, 95% CI = 1.40–2.91; OR = 1.73, 95% CI = 1.19–2.51; respectively) than men with high-income status. Compared to men with high-income status, the OR of women with income-dependent status was the highest in the RUG4 group. The patterns of gender-income effect among cancer and stroke patients were similar, however, the power of the statistical test was much weaker in stroke patients. For example, among stroke patients, the OR of women with income-dependent status and low-income status compared to men with the high-income in RUG4 was 1.48 (95% CI = 0.81–2.69) and 1.34 (95% CI = 0.73–2.46), whereas the OR of their counterparts among cancer patients in RUG4 was 2.01 (95% CI = 1.00–4.02) and 1.76 (95% CI = 0.86–3.58).

## Discussion

The present study demonstrates the pattern of medical care needs in HHC in Taiwan at a national level during 1997 and 2013, which is unique compared to previous studies investigated in a shorter period [8–12]. We found that women with income-dependent status had the highest burden of medical care needs in HHC recipients in Taiwan. The finding is in keeping with the theories of fundamental causes of disease and feminization of poverty [16, 20]. Moreover, the effect of gender-income on medical care needs differs in various diseases.

In general, the highest level of medical care need in HHC was reducing whereas the basic levels of medical care need for HHC are climbing over time in Taiwan during 1998 and 2013. The trends may be explained by the dynamic equilibrium of morbidity hypothesis, suggesting the population who suffer from disease or disability with less severe conditions are increasing due to the longevity [25–30]. The onset of more burdensome symptoms is delayed in those who have already had diseases [30] and the function of vital organs get declined slowly over time [31]. The clinical improvement and healthier behavior, education, and better living environment reduce mortality rate, leading to the reduction of severity of the chronic disease, slowing its rate of progression, and minimizing the effects of severe health problems on mortality but increasing the prevalence of disease [32–35]. Thus, the number of people living with highly-morbid or severe-health conditions gets relatively constant over time, whereas, the prevalence of people with mild and moderate disability or non-lethal impairment increases [29, 36–39].

**Table 4. Multinomial logistic regression model for medical care needs levels in home healthcare (N = 238,176).**

| | RUG2 vs. RUG1 | | RUG3 vs. RUG1 | | RUG4 vs. RUG1 | |
| --- | --- | --- | --- | --- | --- | --- |
| | Adjusted OR[a] (95% CI) | p | Adjusted OR[a] (95% CI) | p | Adjusted OR[a] (95% CI) | p |
| **Age** | 1.03 (1.03–1.04) | <0.0001 | 1.03 (1.02–1.03) | <0.0001 | 0.99 (0.99–0.99) | <0.0001 |
| **Gender** (Ref = male) | 0.93 (0.87–0.99) | 0.0171 | 1.17 (1.10–1.25) | <0.0001 | 1.13 (1.06–1.22) | 0.0004 |
| **Income** (Ref = high) | | | | | | |
| **Dependent** | 1.75 (1.33–2.29) | <0.0001 | 2.34 (1.77–3.09) | <0.0001 | 1.98 (1.44–2.71) | <0.0001 |
| **Low** | 1.67 (1.27–2.19) | 0.0002 | 2.18 (1.65–2.88) | <0.0001 | 1.81 (1.32–2.48) | 0.0002 |
| **Moderate** | 2.09 (1.58–2.76) | <0.0001 | 2.35 (1.77–3.13) | <0.0001 | 1.17 (0.85–1.62) | 0.3349 |
| **Urbanization** (Ref = urban) | | | | | | |
| **Sub-urban** | 0.97 (0.91–1.03) | 0.3525 | 1.02 (0.95–1.08) | 0.6624 | 1.06 (0.99–1.14) | 0.1133 |
| **Sub-rural** | 1.12 (1.00–1.26) | 0.0510 | 1.24 (1.1–1.39) | 0.0004 | 1.26 (1.11–1.43) | 0.0004 |
| **Rural** | 1.10 (0.85–1.42) | 0.4718 | 1.21 (0.93–1.57) | 0.1495 | 1.16 (0.87–1.54) | 0.3066 |
| **Comorbidities** | | | | | | |
| **Cancer** | 0.88 (0.82–0.94) | 0.0004 | 0.70 (0.65–0.75) | <0.0001 | 0.58 (0.53–0.63) | <0.0001 |
| **Neurodegenerative diseases** | 1.78 (1.65–1.92) | <0.0001 | 1.75 (1.62–1.89) | <0.0001 | 1.43 (1.31–1.55) | <0.0001 |
| **Stroke** | 1.90 (1.78–2.03) | <0.0001 | 2.43 (2.27–2.59) | <0.0001 | 2.86 (2.66–3.08) | <0.0001 |
| **Heart failure** | 0.66 (0.62–0.71) | <0.0001 | 0.62 (0.58–0.67) | <0.0001 | 0.62 (0.57–0.67) | <0.0001 |
| **COPD** | 0.71 (0.67–0.76) | <0.0001 | 0.72 (0.68–0.77) | <0.0001 | 0.81 (0.76–0.87) | <0.0001 |
| **Chronic liver disease** | 1.02 (0.94–1.10) | 0.6735 | 0.96 (0.89–1.04) | 0.2787 | 0.85 (0.78–0.93) | 0.0002 |
| **Chronic kidney disease** | 1.16 (1.07–1.26) | 0.0002 | 1.05 (0.97–1.14) | 0.2120 | 0.92 (0.84–1.00) | 0.0620 |
| **Hypertension** | 1.23 (1.13–1.33) | <0.0001 | 1.14 (1.05–1.23) | 0.0014 | 1.00 (0.92–1.09) | 0.9942 |
| **Diabetes** | 1.04 (0.94–1.15) | 0.4669 | 1.12 (1.01–1.24) | 0.0299 | 1.11 (0.99–1.23) | 0.0676 |
| **Coronary artery disease** | 1.03 (0.96–1.10) | 0.3877 | 0.98 (0.91–1.05) | 0.5444 | 0.91 (0.84–0.98) | 0.0092 |
| **Hyperlipidemia** | 0.95 (0.88–1.02) | 0.1307 | 0.86 (0.80–0.92) | <0.0001 | 0.65 (0.60–0.70) | <0.0001 |
| **Atrial fibrillation** | 0.92 (0.84–1.01) | 0.0665 | 0.89 (0.82–0.97) | 0.0106 | 0.92 (0.84–1.02) | 0.0982 |
| **Tuberculosis** | 0.77 (0.70–0.86) | <0.0001 | 0.72 (0.65–0.80) | <0.0001 | 0.76 (0.68–0.85) | <0.0001 |
| **Epilepsy** | 1.35 (1.16–1.57) | 0.0001 | 1.44 (1.24–1.68) | <0.0001 | 1.58 (1.35–1.85) | <0.0001 |

CI = confidence interval, COPD = chronic obstructive pulmonary disease, OR = odds ratio, RUG = resource utilization groups.

[a] Adjusted for all variables in the table.

Despite several medical and social elucidations that have been articulated in attempts to account for the differences in medical care needs for HHC, how gender and income influence this association remains unclear, especially for the older population [10, 12]. In our study, an increasing proportion of women was noted in the groups with higher medical care needs. It could be because women suffer more from nonlethal illness, especially in a higher rate of chronic debilitation disorders [40], and have better longevity than men, leading to their higher medical care needs. In particular, women in the income-dependent group have much higher percentages in the groups with higher medical care needs than women in other income groups, while the proportion of men with income-dependent status low income is lower than men with low income. The findings confirmed our hypotheses 1 and 2 and echoed the theories of fundamental causes of disease and feminization of poverty [16, 20].

Given the remarkable gender and income gaps in medical care needs, the impact of interaction terms of gender and income has been further established in our study. Women with income-dependent status or low-income are significantly associated with a higher level of medical care needs than men with high-income, corroborating Hypothesis 3. Moreover, different diseases may have impacts on the gender-income gaps in medical care needs. For example, similar patterns were found in all study subjects or in cancer patients that women with lower

**Table 5. Multinomial logistic regression model for medical care needs levels in home healthcare focusing on interaction terms of gender and income among all study subjects, cancer patients, and stroke patients.**

| Gender*income (Ref = male*high) | RUG2 vs. RUG1 | | RUG3 vs. RUG1 | | RUG4 vs. RUG1 | |
|---|---|---|---|---|---|---|
| | Adjusted OR[a] (95% CI) | p | Adjusted OR[a] (95% CI) | p | Adjusted OR[a] (95% CI) | p |
| **All study subjects** (N = 238,176) | | | | | | |
| Male*Dependent | 1.68 (1.22–2.32) | 0.0016 | 2.08 (1.50–2.90) | <0.0001 | 1.69 (1.17–2.45) | 0.0050 |
| Male*Low | 1.64 (1.19–2.27) | 0.0024 | 2.00 (1.44–2.78) | <0.0001 | 1.64 (1.13–2.36) | 0.0087 |
| Male*Moderate | 1.88 (1.35–2.61) | 0.0002 | 1.96 (1.40–2.75) | 0.0001 | 1.07 (0.73–1.56) | 0.7390 |
| Female*Dependent | 1.55 (1.13–2.14) | 0.0069 | 2.45 (1.77–3.41) | <0.0001 | 2.02 (1.40–2.91) | 0.0002 |
| Female*Low | 1.43 (1.03–1.98) | 0.0311 | 2.19 (1.57–3.05) | <0.0001 | 1.73 (1.19–2.51) | 0.0040 |
| Female*Moderate | 1.99 (1.42–2.77) | <0.0001 | 2.63 (1.87–3.70) | <0.0001 | 1.14 (0.78–1.67) | 0.4992 |
| Female*High | 0.81 (0.45–1.46) | 0.4787 | 0.75 (0.41–1.39) | 0.3591 | 0.68 (0.34–1.38) | 0.2825 |
| **Cancer patients** (N = 40,266) | | | | | | |
| Male*Dependent | 1.51 (0.87–2.61) | 0.1451 | 1.96 (1.10–3.48) | 0.0218 | 1.78 (0.89–3.56) | 0.1049 |
| Male*Low | 1.52 (0.88–2.63) | 0.1364 | 1.90 (1.07–3.37) | 0.0284 | 1.70 (0.85–3.40) | 0.1364 |
| Male*Moderate | 1.75 (0.99–3.10) | 0.0541 | 1.99 (1.10–3.61) | 0.0230 | 1.23 (0.60–2.52) | 0.5804 |
| Female*Dependent | 1.54 (0.89–2.67) | 0.1238 | 2.37 (1.34–4.20) | 0.0032 | 2.01 (1.00–4.02) | 0.0494 |
| Female*Low | 1.33 (0.75–2.33) | 0.3271 | 2.06 (1.14–3.70) | 0.0163 | 1.76 (0.86–3.58) | 0.1208 |
| Female*Moderate | 2.45 (1.35–4.42) | 0.0030 | 3.45 (1.87–6.38) | <0.0001 | 1.66 (0.79–3.49) | 0.1829 |
| Female*High | 0.53 (0.20–1.40) | 0.1977 | 0.61 (0.22–1.73) | 0.3556 | 0.38 (0.08–1.74) | 0.2130 |
| **Stroke patients** (N = 167,070) | | | | | | |
| Male*Dependent | 1.25 (0.71–2.19) | 0.4462 | 1.49 (0.85–2.63) | 0.1674 | 1.22 (0.67–2.21) | 0.5241 |
| Male*Low | 1.10 (0.62–1.92) | 0.7525 | 1.29 (0.73–2.28) | 0.3782 | 1.08 (0.59–1.96) | 0.8063 |
| Male*Moderate | 1.42 (0.80–2.52) | 0.2369 | 1.41 (0.79–2.52) | 0.2443 | 0.79 (0.43–1.46) | 0.4522 |
| Female*Dependent | 1.10 (0.63–1.93) | 0.7384 | 1.75 (1.00–3.09) | 0.0518 | 1.48 (0.81–2.69) | 0.2012 |
| Female*Low | 1.06 (0.60–1.87) | 0.8468 | 1.64 (0.92–2.92) | 0.0908 | 1.34 (0.73–2.46) | 0.3433 |
| Female*Moderate | 1.43 (0.81–2.55) | 0.2202 | 1.90 (1.06–3.40) | 0.0301 | 0.84 (0.45–1.55) | 0.5660 |
| Female*High | 0.59 (0.22–1.60) | 0.3016 | 0.58 (0.21–1.58) | 0.2862 | 0.56 (0.19–1.66) | 0.2944 |

CI = confidence interval, OR = odds ratio, RUG = resource utilization groups.

[a] Adjusted for age, urbanization, comorbidities.

income status were more likely to receive higher levels of medical care needs compared to their male counterparts. However, the patterns are less significant in stroke patients across the different gender-income status. The difference may be related to the higher prevalence rate and severity of male in stroke patients [41]. More research is needed to further examine the influences of gender and income in disease-specific HHC.

Given the population aging and increased longevity globally, the expanding numbers of the elderly may not only lead to decreasing social benefits per capita but also intensify the health-care cost [3, 42]. The demand for long-term care expenditures and total out-of-pocket spending may exacerbate the inequity due to gender and income. Previous studies showed that people receiving HHC are at high risk of acute hospitalization, emergency room visit, and other acute healthcare utilization, especially for those with high medical care needs [12, 43]. Our findings suggested an increase in public spending for delivering proper preventive, supportive, and therapeutic care for women with income-dependent status or low-income may be crucial. A home-based intervention that integrates health and social care support, or a timely program coordinated with elderly care specialists for maximizing recipients' independence at home may help to reduce the transitions to medical care in higher-cost settings [44, 45].

## Limitations and strengths

Due to the cross-sectional design, the study cannot reveal any causal inferences but only the associations between social determinants and medical care needs. Second, since the NHIRD is a claim-based routinely collected dataset, it does not provide some specific socioeconomic data such as education years, out-of-pocket expense for healthcare, or other economic circumstances. Third, the evaluation of medical care needs was based on the need for special medical care items and on the time-point when patients received HHC for the first time. The change in medical care needs cannot be identified in our study.

## Conclusion

The secular trend of the changing proportions of medical care need levels in HHC in Taiwan fitted the dynamic equilibrium of morbidity hypothesis, put forward by Manton. Lower socioeconomic status can partially explain the higher level of medical care need among women who received HHC in Taiwan, which is consistent with the theories of fundamental causes of disease and feminization of poverty. To reduce the inequity in medical care needs due to gender and income, policymakers could allocate timely integrated care resources for women in lower-income status at home. Further study is needed for exploring the effect of gender-income on medical care needs in different diseases.

## Acknowledgments

We would like to thank Professor Elizabeth L. Sampson and Professor Irene Petersen at University College London for their advice on the study approach and analysis.

## Author Contributions

**Conceptualization:** Fang-Yi Huang, Sang-Ju Yu, Ping-Jen Chen.

**Data curation:** Fang-Yi Huang, Ping-Jen Chen.

**Formal analysis:** Chao A. Hsiung.

**Funding acquisition:** Ping-Jen Chen.

**Investigation:** Ping-Jen Chen.

**Methodology:** Chao A. Hsiung, Ping-Jen Chen.

**Project administration:** Fang-Yi Huang.

**Resources:** Chung-Han Ho, Chao A. Hsiung, Ping-Jen Chen.

**Software:** Chung-Han Ho, Jung-Yu Liao.

**Supervision:** Ping-Jen Chen.

**Validation:** Chung-Han Ho.

**Visualization:** Jung-Yu Liao.

**Writing – original draft:** Fang-Yi Huang, Ping-Jen Chen.

**Writing – review & editing:** Chung-Han Ho, Jung-Yu Liao, Chao A. Hsiung, Sang-Ju Yu, Kai-Ping Zhang, Ping-Jen Chen.

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
