## [Decision Letter · Decision Letter 0]

9 Dec 2020

PONE-D-20-32788

Medical care needs for patients receiving home healthcare in Taiwan: Do gender and income matter?

PLOS ONE

Dear Dr. Chen,

Thank you for submitting your manuscript to PLOS ONE. After careful consideration, we feel that it has merit but does not fully meet PLOS ONE’s publication criteria as it currently stands. Therefore, we invite you to submit a revised version of the manuscript that addresses the points raised during the review process.

We look forward to receiving your revised manuscript.

Kind regards,

Christy Pu

Academic Editor

PLOS ONE

Journal Requirements:

2.We note that you have indicated that data from this study are available upon request. PLOS only allows data to be available upon request if there are legal or ethical restrictions on sharing data publicly. For more information on unacceptable data access restrictions, please see http://journals.plos.org/plosone/s/data-availability#loc-unacceptable-data-access-restrictions.

Reviewers' comments:

Reviewer's Responses to Questions

5. Review Comments to the Author

Reviewer #1: Lines 60-61 - Please rephrase to specify the following:-

1. Was hospitalisation sought for mental healthcare?

2. Was over the counter medication used for mental healthcare?

If not, then the type of conditions for which these services were availed should be mentioned.

Lines 128-130 - Is the use of the term 'factors' methodologically significant here?

If the authors carried out a factor analysis of the independent variables to arrive at the three factors, it should be mentioned and a justification should be given for the same.

If this is a theoretical categorisation of the independent variables, it should be specified.

Lines 131-132 - Please rephrase for clarity.

Please explain in detail how the 'aged-income' groups were created.

Is this is an interaction term? If yes, is it additive or multiplicative? How was this determined?

This should either be explained in detail or a reference should be cited where this technique has been applied.

Line 158 - Please refer to the comment on lines 131-132.

Lines 163-164 - Please rephrase for clarity.

Lines 164-166 - What is the relevance of this sentence here?

Section - Statistical Analysis

There is no mention of descriptive statistics to showcase the distribution of the variables in the population."

Lines 162-163 - Please demonstrate how this was carried out with an example of an independent variable and a mediating and/or a moderating variable.

Table 1 - Please leave this column blank for the disease conditions.

Line 201 - Increasing how? Across time?

Lines 207-208 - This appears to be counter-intuitive, since kidney disease would require more complex care.

Lines 237-238 - What level of medical care needs?

Table 3 - Why are medical care needs not disaggregated by RUG levels?

Line 258 - Please change to income dependent as 'poorer income' can refer to both low-income and dependent individuals.

Lines 254-256 and Table 4 - Please explain the rationale for grouping disease conditions into 'All' and 'Cancer'. Does the category 'All' include cancers?

Table 4 - The title should include 'cancer' since no other disease conditions are being referenced in the table.

Table 4 Note 2 - Where are the p-values shown?

Table 4 - Please regroup columns for clarity. It would be better to group all the RUG levels for all-cause care needs together, and for cancer separately.

Line 289 - Please specify 'income dependent women'.

Lines 289-290 - Rephrase and add information for clarity and correctness.

According to the OR for interaction terms in Table 3, income-dependent women have a higher likelihood of having higher levels of medical care needs, that is, a higher RUG level.

The same is borne out by the results shown in Table 4, where it is seen that the group 'female and income-dependent' has the highest value in every RUG level, for all-cause medical care needs. Similarly, the percentage of female income-dependent individuals shows an increasing trend from RUG1 to RUG2.

Lines 292-293 - Please provide a reference.

Lines 298-299 - Please rephrase for clarity.

Line 314 - Please check this. Should this be similar?

Lines 316-317 - Please rephrase for clarity

Lines 320-322 - Please specify the income groups accurately and uniformly. Throughout the text, the terms used are 'low-income' and 'income-dependent'. 'Poor income' could be incorrectly interpreted as a combination of both.

Lines 325-328 - Please rephrase for clarity.

Lines 347-349 - Please mention why this is essential and what could be the result if such measures are not taken.

Describe in terms of premature mortality, increased out-of-pocket expenditure for households, and costs to the system related to income loss and disability.

---

## [Author Response · Author response to Decision Letter 0]

22 Jan 2021

Response to reviewers’ comments:

Reviewer 1

1. Typo and grammar issues

Author reply: All typo and grammar issues have been revised

2. Lines 60-61 - Please rephrase to specify the following:

a. Was hospitalisation sought for mental healthcare?

b. Was over the counter medication used for mental healthcare?

If not, then the type of conditions for which these services were availed should be mentioned.

Author reply: Thank you for the comment. We have updated the reference and added more content to clearly describe the type of conditions for which these services were availed, and the content mentioned in the revised manuscript and shown below:

Introduction (page 3, line 69-72)

“Some studies found that women are more likely to receive HHC and report greater unmet home care needs than men (8,9), and this gender gap increases by age (10). Yet, other research reported that men presented with higher levels of need for HHC since men had higher rates of most chronic conditions, limitations in activities of daily living, and instrumental activities of daily living than women (11). Moreover, in the patients who received HHC services, men have higher frequencies of medical care utilization than women among the disabled group (12) and also were more likely to have multiple hospitalizations in the last 3 months of life than women in people with dementia (13).”

3. Lines 128-130 - Is the use of the term 'factors' methodologically significant here?

If the authors carried out a factor analysis of the independent variables to arrive at the three factors, it should be mentioned and a justification should be given for the same.

If this is a theoretical categorisation of the independent variables, it should be specified.

Author reply: Thank you for the comment. This is actually a theoretical categorization of the independent variables and we have updated the reference and revised the content. The content mentioned in the revised manuscript is shown below:

Material and methods (page 6, line 133-136)

“According to a behavioral model for explaining health care utilization developed by Andersen and Newman (23), we categorized covariates into three types, including predisposing characteristics (age and gender), enabling characteristics (income and urbanization), and need characteristics (major diseases and comorbidity).” 

4. Lines 131-132 - Please rephrase for clarity.

Please explain in detail how the 'aged-income' groups were created.

Is this is an interaction term? If yes, is it additive or multiplicative? How was this determined?

This should either be explained in detail or a reference should be cited where this technique has been applied.

Author reply: Thank you for the comment. The 'aged-income' was a typo. The correct one is the 'gender-income' group. We have changed the 'aged-income' groups into the 'gender-income' groups. The definition of income levels is mentioned on Page 6 Line 138-144 in the revised manuscript

We have rearranged the sequence of tables to demonstrate our reasoning more clearly. Due to the specific research interest in gender and income, we present the descriptive data of the pattern of medical care needs for subjects categorized in different gender-income groups in Table 3 (page 13). 

Given the significant roles of gender and income levels in medical care needs in multivariate analysis (Table 4), we further generate interaction terms “Gender*Income”, which includes eight gender-income categories, and test their effect in the multinominal regression model (Table 5). The interaction term is multiplicative and the reference group is male with high-income in Table 5.

5. Section - Statistical Analysis

There is no mention of descriptive statistics to showcase the distribution of the variables in the population.

Author reply: Thank you for the comment. We have revised the content, which is mentioned in the revised manuscript and shown below:

Statistical analysis (page 7, line 157-161)

“Descriptive statistics were conducted to display the distribution of the studying variables in the population sample in Table 1. Figure 1 was used to present how medical care needs changing from 1997 to 2013. In addition, χ2 analysis for categorical variables and analysis of variance (ANOVA) test for the continuous variable (age) were used to examine the differences between each independent variable and the level of RUGs.”

6. Line 158 - Please refer to the comment on lines 131-132.

Author reply: Thank you for the comment. The 'aged-income' groups were a typo. The correct one is the 'gender-income' groups. We have changed the 'aged-income' groups into the 'gender-income' groups. The definition of income levels is mentioned on Page 6 Line 138-144 in the revised manuscript

7. Lines 162-163 - Please demonstrate how this was carried out with an example of an independent variable and a mediating and/or a moderating variable.

Lines 163-164 - Please rephrase for clarity.

Lines 164-166 - What is the relevance of this sentence here?

Author reply: Thank you for the comment. We have performed a new multinominal regression test to replace the previous analysis and revised the content, which is mentioned in the revised manuscript and shown below:

Statistical analysis (page 7, line 161-166)

“Because the RUGs were grouped into more than two categories, we adopted multinomial logistic regression to estimate the associations between variables, eight gender-income groups (an interaction term of gender multiplying income), and medical care needs as in Table 4 and Table 5. The analysis was also done for patients with cancer and stroke specifically due to their lowest and highest prevalence in major diseases served by HHC.”

The example of moderating may be shown in the new Table 5 and the revised content is mentioned in the revised manuscript and shown below:

Results (page 15, line 261-267)

”The patterns of gender-income effect among cancer and stroke patients were similar, however, the power of the statistical test was much weaker in stroke patients. For example, among stroke patients, the OR of women with income-dependent status and low-income status compared to men with the high-income in RUG4 was 1.48 (95% CI= 0.81-2.69) and 1.34 (95% CI= 0.73-2.46), whereas the OR of their counterparts among cancer patients in RUG4 was 2.01 (95% CI= 1.00-4.02) and 1.76 (95% CI= 0.86-3.58).”

8. Table 1 - Please leave this column blank for the disease conditions.

Author reply: Thank you for the comment. We have left this column blank for the disease conditions.

9. Line 201 - Increasing how? Across time?

Author reply: Thank you for the comment. We have revised the content, which is mentioned in the revised manuscript and shown below:

Results (page 10, line 203-208)

“The proportion of women are higher than men in groups with higher medical care needs (54.1% versus 45.9% in RUG3, 50.2% versus 49.8% in RUG4), whereas the trend of gender ratio reversed in groups with lower medical care needs. In the group with the highest level of medical care needs, the percentage of people with income-dependent status (54%) is much higher than their counterparts in other medical care needs groups.”

10. Lines 207-208 - This appears to be counter-intuitive, since kidney disease would require more complex care.

Author reply: Thank you for the comment. We have deleted the confusing interpretation in the previous manuscript and focused on the trends of the proportion in each medical care need group among different types of diseases. The trends of the percentages from RUG1 to RUG4 groups are decreasing among major diseases, such as cancer, heart failure, or chronic liver disease, could be interpreted as patients with aforementioned diseases tend to seek medical care in hospital rather than in upgraded home healthcare. The revised content is mentioned in the revised manuscript and shown below:

Results (page 10, line 209-214)

“Trends of the proportion of each medical care need group from RUG1 to RUG4 are dissimilar by the types of diseases. For instance, cancer patients and stroke patients had the lowest and highest percentages in RUG3 (14.93% versus 72.65%) and RUG4 (12.26% versus 75.41%) respectively. In addition, the percentages of cancer patients are decreasing from RUG1 to RUG4 groups (23.20%, 19.08%, 14.93%, and 12.26%) whereas the proportions of stroke patients are increasing (50.86%, 67.93%, 72.65%, and 75.41%) according to Table 2.”

11. Lines 237-238 - What level of medical care needs?

Table 3 - Why are medical care needs not disaggregated by RUG levels?

Author reply: Thank you for the comment. We have replaced multi-ordinal regression models (Table 3 in the previous manuscript) with multi-nominal regression models (Table 4 and 5 in the revised manuscript), and rewritten the whole paragraphs which are mentioned in the revised manuscript and shown below:

Results (page 14-15, line 239-267)

“Multinominal regression analysis for medical care needs

In Table 4, we found that women were more likely to receive RUG 3 (odds ratio, OR= 1.17, 95% confidence interval, CI= 1.10-1.25) and RUG4 (OR= 1.13, 95% CI= 1.06-1.22) than men. Compared to the patients with the high-income status, patients with the income-dependent status were more likely to receive RUG3 (OR= 2.34, 95% CI= 1.77-3.09) and RUG4 (OR= 1.98, 95% CI= 1.44-2.71). Patients with the low-income status (RUG3: OR= 2.18, 95% CI= 1.65-2.88; RUG4: OR= 1.81, 95% CI= 1.32-2.48) had similar patterns in medical care needs with the patients with income-dependent status (RUG3: OR= 2.34, 95% CI= 1.77-3.09; RUG4: OR= 1.98, 95% CI= 1.44-2.71) compared to high-income patients respectively. Moreover, the patterns of medical care needs in HHC differ by types of disease. Particularly speaking, cancer patients were less likely to have higher medical care needs, especially in RUG4 (OR= 0.58, 95% CI= 0.53-0.63). However, stroke patients were most likely to receive HHC in higher medical needs ( RUG3: OR= 2.43, 95% CI= 2.27-2.59; RUG4: OR= 2.86, 95% CI= 2.66-3.08). 

We further examined the associations between interaction terms of gender and income, and RUGs among all study subjects, cancer patients, and stroke patients. As shown in Table 5, among all study subjects, women with income-dependent status and low-income were more likely to receive RUG3 (OR= 2.45, 95% CI= 1.77-3.41; OR= 2.19, 95% CI= 1.57-3.05; respectively) and RUG4 (OR= 2.02, 95% CI= 1.40-2.91; OR= 1.73, 95% CI= 1.19-2.51; respectively) than men with high-income status. Compared to men with high-income status, the OR of women with income-dependent status was the highest in the RUG4 group. The patterns of gender-income effect among cancer and stroke patients were similar, however, the power of the statistical test was much weaker in stroke patients. For example, among stroke patients, the OR of women with income-dependent status and low-income status compared to men with the high-income in RUG4 was 1.48 (95% CI= 0.81-2.69) and 1.34 (95% CI= 0.73-2.46), whereas the OR of their counterparts among cancer patients in RUG4 was 2.01 (95% CI= 1.00-4.02) and 1.76 (95% CI= 0.86-3.58).”

Table 4 (page 16-17)

Table 5 (page 18-19)

12. Lines 254-256 and Table 4 - Please explain the rationale for grouping disease conditions into 'All' and 'Cancer'. Does the category 'All' include cancers?

Author reply: Thank you for the comment. The category 'All' includes cancer and stroke patients. We have added the rationale of doing the analysis not only in all study subjects but also in cancer and stroke patients individually. The revised content is mentioned in the revised manuscript and shown below:

Statistical analysis (page 7, line 164-166)

“The analysis was also done for patients with cancer and stroke specifically due to their lowest and highest prevalence in major diseases served by HHC.”

Results (page 12, line 225-233)

“The relationships between gender, income, and medical care needs 

Table 3 displays the association between gender-income groups and medical care need levels among all study subjects, cancer patients, and stroke patients. We found out there is a significant difference in proportions of gender-income groups among four medical care need levels of HHC. Overall, women with income-dependent status are more likely to have higher medical care needs compared to other gender-income groups. For example, the percentages of women with income-dependent status in RUG1 to RUG4 groups are 26.43%, 26.24%, 30.68%, and 32.07% respectively. At the same time, the percentages of women with high income in RUG1 to RUG4 groups are 0.38%, 0.14%, 0.10%, and 0.15%.”

Table 3 (page 13)

13. Line 258 - Please change to income-dependent as 'poorer income' can refer to both low-income and dependent individuals.

Line 259 - 'However' should be replaced by 'At the same time' because the following finding is not contradictory to the one preceding it.

Author reply: Thank you for the comment. We have changed “poorer income” to “income-dependent status” and also revised the similar confusing words in all the text. We have replaced ‘However’ with ‘At the same time’.

14. Table 4 - The title should include 'cancer' since no other disease conditions are being referenced in the table.

Table 4 - Please regroup columns for clarity. It would be better to group all the RUG levels for all-cause care needs together, and for cancer separately.

Table 4 Note 2 - Where are the p-values shown?

Author reply: Thank you for the comment. We have revised the title and content in the table and moved its sequence to Table 3, which is mentioned in the revised manuscript and shown below:

Table 3 (page 13)

Line 225-233 “Table 3. The relationships between gender, income, and medical care need level in home healthcare among all study subjects, cancer patients, and stroke patients”

Line 236 “All results for χ2 tests between RUGs and eight gender-income groups are all with p values below 0.001.”

15. Line 289 - Please specify 'income dependent women'.

Lines 289-290 - Rephrase and add information for clarity and correctness.

Author reply: Thank you for the comment. We have rewritten the paragraph, which is mentioned in the revised manuscript and shown below:

Discussion (page 20, line 275-281)

“The present study demonstrates the pattern of medical care needs in HHC in Taiwan at a national level during 1997 and 2013, which is unique compared to previous studies investigated in a shorter period (8–12). We found that women with income-dependent status had the highest burden of medical care needs in HHC recipients in Taiwan. The finding is in keeping with the theories of fundamental causes of disease and feminization of poverty (16,20). Moreover, the effect of gender-income on medical care needs differs in various diseases.”

16. Lines 292-293 - Please provide a reference

Author reply: Thank you for the comment. We have provided references in the text.

17. Lines 298-299 - Please rephrase for clarity.

Author reply: Thank you for the comment. We have rephrased the whole paragraph, which is mentioned in the revised manuscript and shown below:

Discussion (page 20, line 282-294)

“In general, the highest level of medical care need in HHC was reducing whereas the basic levels of medical care need for HHC are climbing over time in Taiwan during 2005 and 2013. The trends may be explained by the dynamic equilibrium of morbidity hypothesis, suggesting the population who suffer from disease or disability with less severe conditions are increasing due to the longevity (26–30). The onset of more burdensome symptoms is delayed in those who have already had diseases (30) and the function of vital organs get declined slowly over time (31). The clinical improvement and healthier behavior, education, and better living environment reduce mortality rate, leading to the reduction of severity of the chronic disease, slowing its rate of progression, and minimizing the effects of severe health problems on mortality but increasing the prevalence of disease (32–35). Thus, the number of people living with highly-morbid or severe-health conditions gets relatively constant over time, whereas, the prevalence of people with mild and moderate disability or non-lethal impairment increases (29,36–39).”

18. Line 314 - Please check this. Should this be similar?

Lines 316-317 - Please rephrase for clarity

Lines 320-322 - Please specify the income groups accurately and uniformly. Throughout the text, the terms used are 'low-income' and 'income-dependent'. 'Poor income' could be incorrectly interpreted as a combination of both.

Lines 325-328 - Please rephrase for clarity.

Author reply: Thank you for the comment. We have made the terms 'low-income' and 'income-dependent' consistently and rephrased the whole paragraphs, which is mentioned in the revised manuscript and shown below:

Discussion (page 20-21, line 295-316)

“Despite several medical and social elucidations that have been articulated in attempts to account for the differences in medical care needs for HHC, how gender and income influence this association remains unclear, especially for the older population (10,12). In our study, an increasing proportion of women was noted in the groups with higher medical care needs. It could be because women suffer more from nonlethal illness, especially in a higher rate of chronic debilitation disorders (40), and have better longevity than men, leading to their higher medical care needs. In particular, women in the income-dependent group have much higher percentages in the groups with higher medical care needs than women in other income groups, while the proportion of men with income-dependent status low income is lower than men with low income. The findings confirmed our hypotheses 1 and 2 and echoed the theories of fundamental causes of disease and feminization of poverty (16,20).

Given the remarkable gender and income gaps in medical care needs, the impact of interaction terms of gender and income has been further established in our study. Women with income-dependent status or low-income are significantly associated with a higher level of medical care needs than men with high-income, corroborating Hypothesis 3. Moreover, different diseases may have impacts on the gender-income gaps in medical care needs. For example, similar patterns were found in all study subjects or in cancer patients that women with lower income status were more likely to receive higher levels of medical care needs compared to their male counterparts. However, the patterns are less significant in stroke patients across the different gender-income status. The difference may be related to the higher prevalence rate and severity of male in stroke patients (41). More research is needed to further examine the influences of gender and income in disease-specific HHC.”

19. Lines 347-349 - Please mention why this is essential and what could be the result if such measures are not taken.

Describe in terms of premature mortality, increased out-of-pocket expenditure for households, and costs to the system related to income loss and disability.

Author reply: Thank you for the comment. We have added a paragraph that how the policy change is needed, which is mentioned in the revised manuscript and shown below:

Discussion (page 21-22, line 317-328)

“Given the population aging and increased longevity globally, the expanding numbers of the elderly may not only lead to decreasing social benefits per capita but also intensify the healthcare cost (3,42). The demand for long-term care expenditures and total out-of-pocket spending may exacerbate the inequity due to gender and income. Previous studies showed that people receiving HHC are at high risk of acute hospitalization, emergency room visit, and other acute healthcare utilization, especially for those with high medical care needs (12,43). Our findings suggested an increase in public spending for delivering proper preventive, supportive, and therapeutic care for women with income-dependent status or low-income may be crucial. A home-based intervention that integrates health and social care support, or a timely program coordinated with elderly care specialists for maximizing recipients’ independence at home may help to reduce the transitions to medical care in higher-cost settings (44,45).”

---

## [Editor Report · Decision Letter 1]

10 Feb 2021

Medical care needs for patients receiving home healthcare in Taiwan: do gender and income matter?

PONE-D-20-32788R1

Dear Dr. Chen,

We’re pleased to inform you that your manuscript has been judged scientifically suitable for publication and will be formally accepted for publication once it meets all outstanding technical requirements.

Kind regards,

Christy Pu

Academic Editor

PLOS ONE

---

## [Editor Report · Acceptance letter]

17 Feb 2021

PONE-D-20-32788R1 

Medical care needs for patients receiving home healthcare in Taiwan: do gender and income matter? 

Dear Dr. Chen:

I'm pleased to inform you that your manuscript has been deemed suitable for publication in PLOS ONE. Congratulations! Your manuscript is now with our production department. 

Kind regards, 

on behalf of

Dr. Christy Pu 

Academic Editor

PLOS ONE